

# Protein-protein interaction prediction using enhanced features with spaced conjoint triad and amino acid pairwise distance

Yunus Emre Göktepe

Computer Engineering, Necmettin Erbakan University, Konya, Turkey

## ABSTRACT

Protein-protein interactions (PPIs) are pivotal in cellular processes, influencing a wide range of functions, from metabolism to immune responses. Despite the advancements in experimental techniques for PPI detection, their inherent limitations, such as high false-positive rates and significant resource demands, necessitate the development of computational approaches. This study presents a novel computational model named MFPIC (Multi-Feature Protein Interaction Classifier) for predicting PPIs, integrating enhanced sequence-based features, including a novel spaced conjoint triad (SCT) and amino acid pairwise distance (AAPD), with existing methods such as position-specific scoring matrices (PSSM) and AAindex-based features. The SCT captures complex sequence motifs by considering non-adjacent amino acid interactions, while AAPD provides critical spatial information about amino acid residues within protein sequences. The proposed model was evaluated across three benchmark datasets—*Saccharomyces cerevisiae*, *Helicobacter pylori*, and human proteins—demonstrating superior performance in comparison to state-of-the-art models. The results underscore the efficacy of integrating diverse and complementary features, achieving significant improvements in predictive accuracy, with the model achieving 95.90%, 99.33%, and 90.95% accuracy on the *Saccharomyces cerevisiae*, *Helicobacter pylori*, and human dataset, respectively. This approach not only enhances our understanding of PPI mechanisms but also offers valuable insights for the development of targeted therapeutic strategies.

## INTRODUCTION

Proteins are fundamental macromolecules that play critical roles in cellular processes. They are essential for cellular health and function, impacting nearly every aspect of life since their diverse functions contribute to growth, metabolism, structural support, and immune responses. Proteins perform most of these functions by interacting with each other and breaking these interactions. Deciphering these interactions represents a critical step towards unraveling the intricacies of cellular function and paving the way for the design of targeted therapies (*Zhou & Shan, 2001*).

Corresponding author
Yunus Emre Göktepe,
yegoktepe@gmail.com

These interactions occur as an outcome of some structural forces or physicochemical events like hydrophobicity or hydrogen bonding. It is essential to comprehend connections in the basis of molecules to reveal biological functions (*Sahni et al., 2023*). Accordingly, the protein-protein interaction (PPI) prediction process is very important and is being intensively researched.

PPIs are grouped in a number of types according to their basic properties and structural forms. Based on their affinity PPIs categorized as obligate and non-obligate. In an obligate PPI the elements of the interaction cannot exist particularly. A non-obligate interaction means that the elements are independent and stable in their environment. They are analyzed as homooligomeric or heterooligomeric based on their organization. A PPI is labelled as homooligomeric if the interaction takes place among similar residues otherwise it is called as heterooligomeric. Lastly, PPIs categorized as transient and permanent according to duration of interaction. The lifetime of transient interactions ends with the separation of proteins again after a certain period of time. But the latter ones are steady with their new form (*Acuner Ozbabacan et al., 2011*).

Several experimental methodologies are employed by the scientists to determine the interactions of proteins. Yeast two-hybrid is a common technique used to detect PPIs which involves the reconstitution of a transcriptional regulator within yeast through the interaction of two proteins (*Ito et al., 2001*). Protein chips are surfaces containing immobilized proteins which can be utilized to test for PPIs (*Zhu & Snyder, 2003*). The tandem affinity purification method involves creating fusion proteins with a dual-affinity tag, allowing for two consecutive purification steps in order to identify interacting proteins by mass spectrometry (*Puig et al., 2001*). Immunoprecipitation is a technique which uses antibodies to pull down a protein from a solution, along with its interacting partners (*Niranjanakumari et al., 2002*).

The emergence of high-throughput methodologies has facilitated the investigation of thousands of pairwise interactions, thereby enhancing our understanding of the interactome, the comprehensive network of physical interactions within a cell or organism. High-throughput technologies have indeed revolutionized PPI prediction, but they also come with several drawbacks. These approaches often suffer from a high false-positive rate. This implies that numerous anticipated interactions might not manifest in practice, leading to inaccurate results. Some potential interactions might be missed as these technologies do not cover all possible interactions due to their inherent limitations. Moreover, experimental methods for PPI detection can be costly, labor-intensive, and limited in scalability, making them less accessible for some researchers (*Li et al., 2021*). The inherent limitations of experimental techniques necessitate the creation of computational approaches to predict PPIs (*Ding & Kihara, 2019*).

In the field of modern biology and medicine, computational methods for predicting PPIs have arisen as an indispensable tool. These methods serve as a complementary approach to experimental detection techniques, offering valuable insights into the intricate network of protein interactions (*Behbahani, Mohabatkar & Nosrati, 2016*).

Computational approaches for PPI prediction provide efficient and scalable alternatives to experimental techniques, enabling the rapid prediction of potential interactions

(*Meher et al., 2017*). These methods leverage various biological data types, which can be broadly categorized into structure-based, network-based, and sequence-based approaches.

• Structure-based methods: These methods utilize three-dimensional protein structures to predict binding affinities and interaction interfaces (*Agrawal, Helk & Trout, 2014*; *Zhang et al., 2012*). However, their applicability is limited due to the lack of experimentally resolved structures for many proteins.

• Network-based methods: These approaches analyze existing PPI networks and databases, predicting new interactions based on topological and functional similarity. While powerful, they rely heavily on the quality and completeness of existing network data (*Feng et al., 2024*; *Gong et al., 2024*; *Hu et al., 2021*; *Kovács et al., 2019*; *Zeng et al., 2024*).

• Sequence-based methods: By using primary amino acid sequences, these methods identify interaction motifs and domains through statistical and machine learning techniques (*Göktepe, İlhan & Kahramanlı, 2016*; *Jia et al., 2019*). Sequence-based methods are particularly advantageous for high-throughput analyses, as they do not require structural data and can predict interactions for a wide range of proteins, including those with unresolved structures (*Göktepe & Kodaz, 2018*; *Pan et al., 2021*).

Sequence-based approaches to PPI prediction are crucial due to their ability to utilize the primary structure of proteins for interaction prediction. These methods leverage amino acid sequences to identify conserved motifs and interaction domains, which are fundamental for binding. The broad applicability of sequence-based approaches facilitates the examination of a vast number of proteins, including those without resolved structural information, thus providing insights into the interactome. Sequence-based methods benefit significantly from advancements in machine learning algorithms, which can detect intricate patterns within extensive datasets, thereby enhancing prediction accuracy. Additionally, the integration of evolutionary information through sequence alignment further improves these models by highlighting functionally relevant interactions conserved across different species. Consequently, sequence-based approaches are indispensable in the high-throughput prediction of PPIs, facilitating the annotation of interactomes and advancing our understanding of cellular processes (*Pan et al., 2021*; *Shen et al., 2007*; *Zhou & Shan, 2001*).

Despite their advantages, existing computational approaches face several limitations. Structure-based methods are constrained by the scarcity of structural data, while network-based methods depend heavily on existing interaction databases, which may not be comprehensive. Sequence-based methods, while versatile, often struggle with modeling complex interaction patterns and may require extensive feature engineering to achieve optimal performance.

To address these limitations, this study introduces a novel computational framework, MFPIC (Multi-Feature Protein Interaction Classifier) for predicting PPIs by integrating diverse advanced features and machine learning techniques. The primary features used in this model include conjoint triads and a novel variant of it called the spaced conjoint triads, PSSM (Position-Specific Scoring Matrix), AAindex-based features, hydrophobicity, hydrophilicity, and amino acid pairwise distance (AAPD). The conjoint triad method captures the cognizance of sequential arrangement of amino acids, while spaced conjoint

triad variant extends this by considering triplets of amino acids with possible gaps between them, thereby capturing more complex interactions. PSSM features add evolutionary information by considering the likelihood of each amino acid's occurrence in a given position, and AAindex features provide a comprehensive set of physicochemical properties of them. Hydrophobicity and hydrophilicity measures offer insights into the protein's interaction with its environment, while the AAPD feature calculates the pairwise distances between amino acids in a sequence, capturing spatial information that is critical for understanding protein structure and interaction dynamics. These diverse features were combined to create a robust feature set that was then used to train a deep learning model.

This article introduces and evaluates an innovative model developed for predicting PPIs. Following the introduction, the second section discusses the materials and methods, detailing the datasets used, feature extraction techniques, and the structure of the deep learning-based model. The third section presents and discusses the performance measurements obtained from experiments conducted on datasets from *Saccharomyces cerevisiae*, human, and *Helicobacter pylori* proteins. Finally, the fourth section summarizes the results, emphasizes the advantages of the proposed model, and offers suggestions for future research.

## MATERIALS AND METHODS

In this section, the whole process of accurately predicting PPIs is described meticulously. Specifically, the problem of PPI prediction is defined first. Then, the data is prepared and six enhanced features are selected to form the feature set. Lastly, a detailed explanation is provided for each element of the MFPIC architecture.

### Problem definition

Predicting PPIs can be approached as a two-class classification issue. This research introduces this problem through a mapping function $f : X \rightarrow Y$, where $X$ is the set of input features derived from protein sequences, and $Y$ is the set of predicted interaction classes.

Given a pair of protein sequences $P_1 = \left(P_{1_1}, P_{1_2}, \ldots, P_{1_n}\right)$ and $P_2 = \left(P_{2_1}, P_{2_2}, \ldots, P_{2_n}\right)$ where $P_{1_i}$ and $P_{2_j}$ imply the $i$-th and $j$-th amino acid residues in sequences $P_1$ and $P_2$, respectively. Our goal is to ascertain if these two proteins interact.

The prediction task is defined by the function $f$, where:

$$f(P_1, P_2) = y.$$

Here, $y \in \{0, 1\}$ is the predicted class for the protein pair, with $y = 1$ indicating that the proteins interact, and $y = 0$ indicating no interaction.

To accurately predict $y$, a set of features $X = (x_1, x_2, \ldots, x_k)$ is extracted from the protein sequences $P_1$ and $P_2$. These features include traditional representations like conjoint triads and position-specific scoring matrix (PSSM) profiles, besides novel features such as spaced conjoint triads (SCT) and amino acid pairwise distance (AAPD). The classification model is then trained on these features to learn the mapping function $f$, enabling it to anticipate the likelihood of interaction between specific protein pairs.

The prediction of PPIs considered here is a binary classification task. The positive class, indicating an interaction, can be labeled with "+" and the negative class, indicating an interaction, can be labeled with "−". For the dataset used, the complete dataset can be denoted by $DS = DS_+ \cup DS_-$, where $DS_+$ and $DS_-$ are the parts of the dataset containing positive and negative examples, respectively.

## Data collection

In this study, datasets from different organisms such as *Saccharomyces cerevisiae*, *Helicobacter pylori*, and human were used for the prediction of PPIs. These datasets contain PPI data that are widely used in the relevant literature and have high reliability. Efficacy of the designed model could be assessed extensively using these databases, which are frequently preferred by researchers.

The core dataset for *Saccharomyces cerevisiae*, initially exploited by *Guo et al. (2008)*, comprises 11,188 pairs of proteins, including 5,594 positive protein pairs and an equal number of negatives, which were gathered from diverse sources to achieve a balanced comparison. This dataset is obtained from DIP and provides essential insights into the fundamental functions of the *Saccharomyces cerevisiae* cell and for exploring protein interaction networks (*Xenarios et al., 2002*).

*Helicobacter pylori* was utilized in this research based on the dataset employed in the research conducted by *Martin, Roe & Faulon (2005)*. *Helicobacter pylori* is a bacterium that has been associated with various gastrointestinal diseases in humans, such as gastritis, peptic ulcers, and stomach cancer. Studies on protein interactions of this bacterium are important for understanding disease pathogenesis and identifying potential therapeutic targets. The study of *Martin, Roe & Faulon (2005)* contains reliable PPI data for this bacterium and this dataset was used as a reference in this study. There are a total of 2,916 protein pairs in the *Helicobacter pylori* dataset, 1,458 of which are positive and 1,458 are negative.

The human dataset utilized in this research was obtained from the Human Protein Reference Database (HPRD) and utilized in the work of *Huang et al. (2015)*. An overall number of 8,161 protein pairs were included in this dataset, of which 3,899 are positive and 4,262 are negative. HPRD is a comprehensive database of the human proteome, providing detailed information on PPIs, protein functions, and post-translational modifications (*Keshava Prasad et al., 2009*). The human dataset is of critical importance in biomedical researches, particularly to enlighten the pathogenesis of diseases and designing therapeutic strategies. Table 1 summarizes the details of the benchmark datasets used.

The AAindex database is an extensive repository that catalogs numerical indices related to the diverse physicochemical and biochemical characteristics of amino acids and pairs of them. It is structured with three sections: AAindex1, which includes indices for individual amino acids based on 20 numerical values, AAindex2, which contains amino acid substitution matrices, and AAindex3, which focuses on statistical potentials related to protein contacts. These indices are compiled from issued researches and are extensively utilized in bioinformatics for tasks to analyze and predict structures and functions of proteins. The database is updated regularly and is accessible online through the

**Table 1 Details of the benchmark datasets.**

| Dataset | Positive pairs | Negative pairs | Total PPI data |
|---|---|---|---|
| *Saccharomyces cerevisiae* | 5,594 | 5,594 | 11,188 |
| *Helicobacter Pylori* | 1,458 | 1,458 | 2,916 |
| Human | 3,899 | 4,262 | 8,161 |

GenomeNet at https://www.genome.jp/ftp/db/community/aaindex/aaindex1 for local use (*Kanehisa, 1997*). The AAindex database provides a valuable resource for researchers seeking to understand the complex relationships between amino acid properties and protein behavior, contributing to advancements in fields including subcellular localizations proteins, immunogenicity estimation, and PPI studies (*Kawashima et al., 2008*).

## Representation of proteins

The ability to accurately predict PPIs is contingent upon the quality of features extracted from protein sequences. These features essentially capture the sequence information that dictates interaction potential. Various feature extraction techniques have been explored to translate this sequence data into usable features. However, the ideal approach should not only unveil properties governing protein interactions but also address limitations of previous methods, particularly those related to protein length variation.

A comprehensive set of features is extracted from the amino acid sequences to capture diverse aspects of protein interactions.

### Conjoint triads

The Conjoint Triad encoding method, introduced by *Shen et al. (2007)*, aims to enhance the prediction of PPIs. In this method, protein sequences are represented as vector spaces which captures essential features of amino acids. Specifically, the vector space is constructed by grouping the 20 naturally present amino acids according to their dipoles and side chain volumes. As a result, all protein sequences are transformed to 343-dimensional feature vectors, providing valuable information for predicting interactions between proteins (*Shen et al., 2007*).

The construction of the vector space is achieved by grouping the 20 naturally occurring amino acids according to their dipole moments and the volumes of their side chains. Consequently, all protein sequences are converted into a 343-dimensional feature vector, which serves as a crucial source of information for the prediction of PPIs.

Conjoint triads capture the sequence order information of amino acids by considering all possible triplets of amino acids within a given sequence. For each protein sequence of both interacting proteins, the frequency of occurrence for each triplet is calculated.

Let $P = \{A_1, A_2, \ldots, A_n\}$ be a protein sequence of length $n$, where each $Ai$ is an amino acid in the sequence. The number of possible triads T in the sequence is given by:

$$T = \{A_i, \ A_{i+1}, \ A_{i+2} \ | 1 \le i \le n - 2\}$$

The formula for the frequency count of a triad $t = A_i, \ A_{i+1}, \ A_{i+2}$ in a sequence of length $n$ is:

$$f(A_i, \ A_{i+1}, \ A_{i+2}) = \frac{number \ of \ occurences \ of \ (A_i, \ A_{i+1}, \ A_{i+2})}{n - 2}$$

where $Ai, Ai + 1$, and $Ai + 2$ represent different amino acids and $n - 2$ is the total number of possible overlapping triplets in the sequence.

### Spaced conjoint triads

To enhance the model's ability to grab more intricate interactions, the SCT is suggested. This feature considers triplets of amino acids with possible gaps between them. For example, in the sequence 'MVICL', the spaced conjoint triad 'MIL' can be identified. The formula to count a spaced triad $(A_i, A_j, A_k)$ with a maximum gap size $g$ is:

$$f_{spaced}(A_i, \ A_j, \ A_k) = \frac{number \ of \ occurences \ of \ (A_i, \ *, \ A_j, \ *, \ A_k)}{n - 2 - g}$$

where * denotes any amino acid. This method improves prediction accuracy by considering interactions between spatially close but sequentially distant amino acid residues in protein sequences. Pseudocode for extracting spaced conjoint triads feature is given Algorithm 1.

Figure 1 schematically illustrates the feature extraction process using the conjoint triad method and highlights the difference between the standard conjoint triad and spaced conjoint triad approaches.

Figure 1. The feature extraction process using the standard conjoint triad *vs* spaced conjoint triad approaches.

SCT was developed as an extension of the classical conjoint triad method to capture spaced motifs in protein sequences. This method enables modeling of non-orderly but spatially related amino acid interactions. For example, while the standard conjoint triad method only takes into account consecutive triplet amino acid groups, SCT represents sequence motifs more comprehensively by also considering amino acid groups spaced a certain distance apart. This feature of SCT is critical, especially in capturing indirect binding motifs that determine protein-protein interactions.

### Position-specific scoring matrix

The PSSM features were computed through enumerating the occurrences of each amino acid at each position across all sequences. The resulting frequency matrix was converted to log-odds scores to produce a PSSM vector for each protein sequence, accounting for sequence length variability (*Jones, 1999*).

---

**Algorithm 1** Pseudocode for extracting spaced conjoint triads.

Input: Dataset of protein sequences

Output: feature list of spaced triad counts

Begin

    Define all possible triads (triplets of amino acids).

    Initialize a dictionary to store the counts of each spaced triad.

    For each protein sequence in the dataset:

        Initialize a dictionary to count spaced triads for this sequence.

        Slide a window of size 5 across the sequence:

            Extract the first, third, and fifth amino acids in the current window.

            Form a spaced triad using these amino acids.

            Increment the count for this spaced triad in the dictionary.

        Divide each triad count by the sequence's total spaced triad count to normalize the values.

        Append the normalized spaced triad counts to the feature list.

    Return the feature list containing the normalized spaced triad counts.

End

---

PSSM scores are derived from sequence alignments of homologous proteins and are used to estimate the propensity of every amino acid residing at a given site. The PSSM score for amino acid $A_i$ at position $j$ is:

$$PSSM_{i,j} = log \frac{P(A_i \ at \ j \ |alignment)}{P(A_i)}$$

where $P(A_i \ at \ j \ |alignment)$ is the probability of observing $A_i$ at position $j$ in the alignment, and $P(A_i)$ is the overall likelihood of $A_i$ in the dataset. The pseudocode of the method used to extract PSSM features is presented in Algorithm 2.

### AAindex features

AAindex provides a broad range of physicochemical properties for amino acids (*Kawashima et al., 2008*). AAindex features are extracted based on the dataset provided, which include properties such as hydrophobicity, hydrophilicity, polarity, and molecular volume. Each amino acid in the sequence is characterized by these properties, and the features are averaged across the sequence.

### Hydropathy

Hydropathy is a commonly used term to encompass both hydrophobicity and hydrophilicity (*Biro, 2006*). These two features measure the tendency of amino acids to interact with water or nonpolar environments. These features are calculated using established scales such as the Kyte-Doolittle measure for hydrophobicity. The Kyte-Doolittle measure is a well-established hydrophobicity scale that has been extensively utilized to pinpoint hydrophobic residues within protein sequences. Positive scores on this

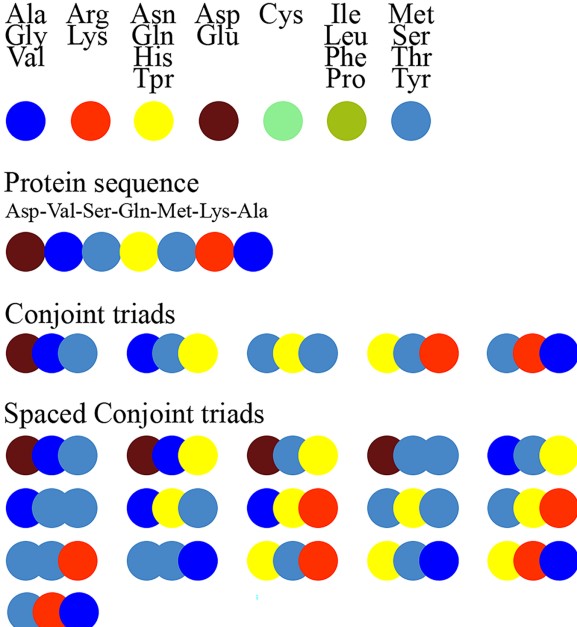

**Figure 1** **Extraction of conjoint triads and spaced conjoint triads.**

scale signify hydrophobic amino acids, suggesting potential localization in the protein's interior. Notably, the scale's sensitivity to window size allows for the identification of both surface-exposed residues and transmembrane domains. By adjusting the window size, researchers can tailor the scale to detect hydrophobic regions of varying lengths, providing valuable knowledge about protein structure and function. The formulation of this scale is given below (*Kyte & Doolittle, 1982*):

$$Hydropathy = \sum_{i=1}^{s} \frac{1}{s} Hydropathy(A_i)$$

where s denotes the total number of amino acids in the sequence, and $Hydropathy(A_i)$ expresses the hydropathy value of amino acid $Ai$ on the Kyte-Doolittle scale.

### Amino acid pairwise distance

AAPD is a feature that aims to capture the spatial relationships in the three-dimensional structures of proteins by calculating the distances between amino acid pairs in protein sequences. While traditional methods focus on the sequence order, AAPD goes beyond this order and takes into account the structural context of proteins. For example, the distances between key amino acids in the active sites of proteins play an important role in determining protein-protein interactions. By analyzing these distances, AAPD provides a deeper understanding of the binding potential of proteins.

AAPD captures the spatial relationships between amino acids within a sequence. It is computed as the distance between every pair of amino acids in the sequence. For amino acids $Ai$ and $Aj$ at positions p$i$ and p$j$, the pairwise distance is:

---

> **Algorithm 2 Pseudocode for extracting PSSM features.**
>
> Input: Dataset of protein sequences
>
> Output: feature list of PSSM vectors
>
> Begin
>
>   Foreach protein sequence in the dataset:
>
>     Form a PSSM for the sequence.
>
>     Foreach position in the sequence:
>
>       Record the score for each amino acid.
>
>     Flatten the PSSM to a feature vector.
>
>     Append the PSSM feature vector to the feature list.
>
>   Return the feature list containing the PSSM feature vectors.
>
> End

$$Distance\left(A_i,\ A_j\right) = \left|p_i - p_j\right|$$

where $|pi - pj|$ is the absolute difference in their positions. And the average distance between pairs of amino acids of the same type in the protein sequence is calculated by the following formula:

$$AAPD = \frac{1}{C} * \sum_{i=1}^{c} distance\left(A_i,\ A_j\right)$$

where C denotes the total number of amino acid pairs of the same type.

AAPD is a feature extraction technique that focuses on calculating the distance between a specific pair of amino acids in a sequence. With determining the physical or positional distances between key amino acid residues, AAPD can capture spatial relationships and structural properties of the protein that are critical for understanding PPIs. This feature is particularly useful in identifying how certain regions of a protein may contribute to its interaction capabilities, enhancing the capacity of the model to predict interactions rooted in sequence data. The pseudocode shown in Algorithm 3 describes the method used to extract AAPD features.

## Multi-feature protein interaction classifier model

A deep learning model was built employing extracted features. The visual representation of the presented model given in Fig. 2, clearly represents the systematic approach undertaken by the proposed model. By integrating various feature extraction methods, including a novel conjoint triad variant and amino acid pairwise distance, the model is designed to capture intricate patterns within protein sequences that are indicative of interactions. The deep learning architecture further enhances the model's capacity to learn complex relationships, leading to robust and accurate predictions.

| Algorithm 3 Pseudocode for extracting aapd features. |
| --- |
| Input: Dataset of protein sequences |
| Output: AAPD feature list |
| Begin |
|    Initialize an empty list for AAPD features. |
|    Foreach protein sequence in the dataset: |
|       Initialize an empty list for storing distances. |
|       Foreach amino acid in the sequence: |
|          Find all positions of it in the sequence. |
|          Calculate the pairwise distances between each occurrence. |
|          Append the distances to the list. |
|       Calculate the mean of the distances for the sequence. |
|       Append the mean distance to the AAPD feature list. |
|    Return the AAPD feature list. |
| End |

Figure 2 shows the flowchart of the model. This diagram shows the flow of the proposed model for protein-protein interaction prediction. In the first stage, meaningful features (*e.g.*, conjoint triads and PSSM) are extracted from amino acid sequences and transferred to the input of the model. The input features are learned by passing them through multiple dense layers, and activation functions (LeakyReLU), batch normalization and dropout are applied in this process. The model finally provides protein interaction prediction by passing the learned features to the classifier layer.

The model structure includes several dense layers, incorporating batch normalization to stabilize learning and reduce internal covariate shift. LeakyReLU activation functions are used to introduce non-linearity, and dropout layers are included to prevent overfitting.

The model is set up using the Adam optimizer, which adapts the learning rate based on the first and second moments of the gradients. Early stopping and learning rate reduction are utilized to preclude overfitting and to yield optimal convergence during training. Model checkpointing is employed to preserve the optimal model configuration throughout the training process.

The model architecture employed in this study is a fully connected deep neural network with the configuration described below. The input layer of the model consists of 38,892 nodes, reflecting the dimensionality of the feature vector. The model involves four hidden layers, all utilizing batch normalization and LeakyReLU activation, featuring an alpha parameter of 0.1 to avoid the dying ReLU problem. Dropout with a rate of 0.3 is applied after each hidden layer to prevent overfitting.

The hidden layers are structured with 512 nodes in the first layer, 256 nodes in the second layer, 128 nodes in the third layer, and 64 nodes in the fourth layer. The output

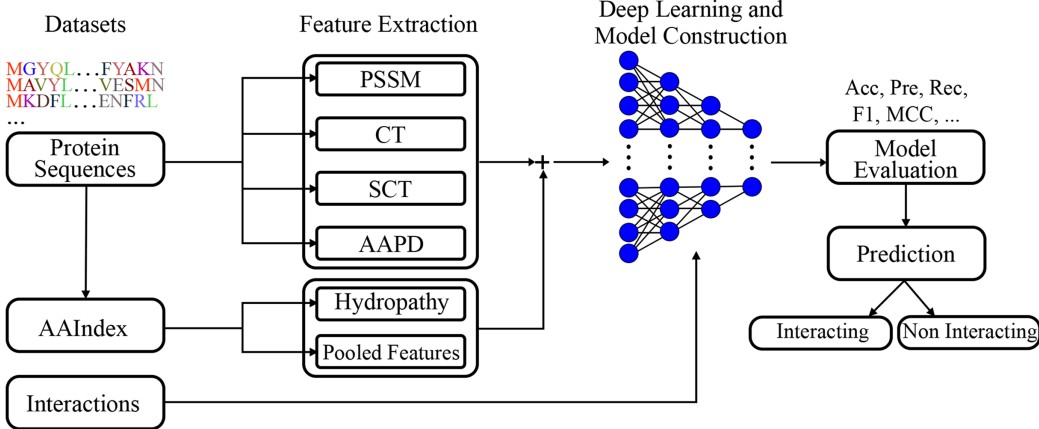

**Figure 2 Visual representation of the proposed model.**

layer comprises a single node, utilizing a sigmoid activation function to generate a probability score that indicates the likelihood of PPI.

The overall pseudocode of the proposed model is presented in Algorithm 4.

The model was trained employing a cross-validation method to assess its generalizability. K-fold cross-validation was used in the evaluation of the model. In our study, 10-fold cross-validation was preferred considering the sizes and distributions of the data sets. This method involves dividing the data set into 10 equally sized subgroups. In each iteration, one subgroup was used as the test set, while the remaining nine subgroups were used for training. This process ensures that the model is tested on the entire data and the performance results are averaged.

The data separation ratios used in cross-validation are as follows:

Training Set: 90% of the data set.

Test Set: 10% of the data set.

This procedure was applied by taking care to ensure that each subgroup represents its classes proportionally in order to prevent imbalance between classes (stratified k-fold).

In addition, the sizes and class distributions of the data sets were also taken into account to ensure the consistency of the model. This method aims to fairly measure the performance of the model on different data subsets and reduce possible variations.

The feature extraction and model training processes were implemented in Python using libraries such as TensorFlow and scikit-learn.

## Performance evaluation

A set of critical measures, including accuracy, precision, recall, F1-score, and Matthews correlation coefficient (MCC) were used to evaluate the efficiency of proposed PPI model. The percentage of properly identified cases is measured by accuracy, which gives an overall impression of the model's performance. By calculating the ratio of real positive outcomes within all positive predictions, precision, alternatively referred as positive predictive value (PPV), highlights the model's accuracy in forecasting interactions while reducing false positives. Recall, sometimes called as the true positive rate (TPR), measures the model's

**Algorithm 4 Pseudocode for multi-feature protein interaction classifier (MFPIC) model.**

Input: Dataset D

Output: Dataset D with new feature 'duration_difference'

**Begin**

    Load the protein sequence data.

    Perform data preprocessing:

        Clean the sequences.

        Encode the sequences if necessary.

    Extract features:

        Compute Conjoint Triad features.

        Compute Spaced Conjoint Triad features.

        Compute PSSM features.

        Compute *AAindex* features.

        Compute Hydrophobicity and Hydrophilicity features.

        Compute AAPD features.

    Combine all extracted features into a single feature matrix.

    Partition the dataset into distinct training and testing subsets.

    Standardize the features.

    Define the deep learning model architecture:

        Input layer.

        Dense layers with Batch Normalization, Leaky ReLU activation, and Dropout regularization.

        Output layer with sigmoid activation.

    Compile the model using the Adam optimizer and binary cross-entropy loss function.

    Train the model using Early Stopping, Learning Rate Reduction, and Model Check points.

    Evaluate the model on the test set.

    Make predictions and assess the performance.

**End**

potential to detect positive occurrences like genuine interactions. Because it takes into account both false positives and negatives, the F1 measure, the harmonic mean of precision and sensitivity, offers a balanced assessment and is especially helpful in situations when datasets are unbalanced. Lastly, a reliable indicator of the model's prediction ability is provided by the MCC, which offers a thorough metric that considers the TP, TN, FP, and FN quadrants of the confusion matrix. When combined, these measures ensure a thorough picture of the model's effectiveness, ensuring that it reduces missed and incorrect detections while also reliably detecting interactions.

$$ACC = \frac{TP + TN}{TP + TN + FP + FN}$$

$$Precision = \frac{TP}{TP + FP}$$

$$Recall = \frac{TP}{TP + FN}$$

$$F1\ Score = \frac{2*Precison*Recall}{Precision + Recall}$$

$$MCC = \frac{TP*TN - FP*FN}{\sqrt{(TP + TN)*(TP + FP)*(TN + FP)*(TN + FN)}}$$

These formulas are based on the values of TP, TN, FP, and FN. TP (true positives) reflects the number of correctly identified interacting pairs, while TN (true negatives) denotes the number of correctly identified non-interacting pairs. FP (false positives) is the number of protein pairs that are incorrectly identified as interacting, and FN (false negatives) represents the number of protein pairs that are incorrectly identified as non-interacting. The MCC accounts for both overestimations and underestimations in predictions, providing a comprehensive performance measure. MCC yields a value ranging from −1 to +1, where −1 points a total discrepancy among the predictions and the observed outcomes, 0 points a prediction that is roughly random, and +1 signifies a perfect prediction of PPIs.

For clarity, the PPI prediction model developed in this research is named as MFPIC and the prediction generation process is as visualized in Fig. 2. The source codes of the model and the datasets can be found at https://github.com/yegoktepe/predictPPI (*Göktepe, 2025*).

## RESULTS AND DISCUSSIONS

In this section the performance of the MFPIC estimator model, developed for PPI prediction, was evaluated through experiments that adhered to the procedures characterized in "Materials and Methods".

The effectiveness of the developed model to predict PPIs was rigorously gauged across three datasets: *Saccharomyces cerevisiae*, human, and *Helicobacter pylori* proteins. These datasets, frequently utilized in PPI prediction literature, provide a robust foundation for assessing model performance.

### Performance of the MFPIC

The primary goal of this study was to enhance PPI prediction accuracy by integrating a novel set of features, including SCT and AAPD, with established methods like conjoint triads, PSSM profiles, and AAindex features. The results demonstrated a substantial increase in predictive accuracy, particularly by combining the SCT and AAPD features.

For the *Saccharomyces cerevisiae* dataset, the proposed model acquired an accuracy of 95.90%, a precision of 98.34%, a recall of 93.37%, a specificity of 98.43%, an F1 score of 95.79%, and an MCC score of 91.91%. These results emphasize the effectiveness of the SCT and AAPD features in capturing complex interaction patterns that were not fully

accounted for by traditional features alone. For the human dataset, the model achieved experimental results of 99.33%, 99.11%, 99.49%, 99.18%, 99.30%, and 98.65% for accuracy, precision, recall, specificity, F1, and MCC measures, respectively. The performance on the *Helicobacter pylori* dataset was similarly impressive with 90.95% accuracy and 90.88% recall. For the precision, specificity, F1, and MCC criteria, values of 91.00%, 91.02%, 90.94%, and 81.89% were achieved, respectively. This improvement demonstrates that the proposed model is particularly adept at identifying interactions in more challenging datasets where traditional methods often struggle. The results obtained by the proposed model on *Saccharomyces cerevisiae*, human and *Helicobacter pylori* datasets are as shown in Table 2.

For the *Saccharomyces cerevisiae* dataset, the calculated FPR and FNR are 6.31% and 1.65%, respectively. Similarly, for the human dataset, the FPR is 0.47%, while the FNR is 0.90%. For the *Helicobacter pylori* dataset, the FPR is 9.10%, and the FNR is 8.99%. These metrics illustrate dataset-specific challenges. For instance, the relatively higher FPR and FNR in the *Helicobacter pylori* dataset can be attributed to the noisier and potentially less distinguishable patterns in this dataset.

False positives (FP) indicate cases where negative samples are incorrectly classified as positive. A high FPR may suggest overlap in feature spaces between the two classes or suboptimal feature extraction. Reducing FPR is particularly crucial in scenarios where false alarms carry significant consequences, such as in high-throughput biological screening. Our future efforts will include exploring advanced feature selection and dimensionality reduction techniques to better separate the feature spaces of positive and negative samples.

False negatives (FN), on the other hand, reflect positive samples misclassified as negative. The FNR is a critical metric in applications where missing true positives has severe implications, such as identifying disease-related proteins. The observed FNR indicates that certain true positive samples may exhibit features that deviate from typical patterns, possibly due to biological variability. Enhancing model sensitivity through strategies such as incorporating additional training data or employing ensemble methods could help mitigate this limitation.

Additionally, the balance between FPR and FNR highlights the inherent trade-offs in classification models. For datasets like *Saccharomyces cerevisiae* and human, where the FPR and FNR are relatively low, the model demonstrates robust performance with minimal misclassifications. However, the higher FPR and FNR in the *Helicobacter pylori* dataset suggest that dataset-specific preprocessing (*e.g.*, noise filtering or data augmentation) and tailored model architectures could further enhance performance.

## Comparative analysis with previous methods

To further validate the efficacy of the proposed model, a performance comparison was conducted *vs.* several state-of-the-art PPI prediction models. These models included sequence-based approaches, structure-based methods, and integrated techniques that combine multiple data sources. The comparative analysis demonstrated that the proposed method outperformed these models across all datasets. The comparisons are summarized in Table 3–5 for the *Saccharomyces cerevisiae*, human, and *Helicobacter pylori* datasets,

**Table 2 Experimental results of proposed model for *Saccharomyces cerevisiae*, human, and *Helicobacter pylori* datasets.**

| Dataset | *Saccharomyces cerevisiae* | Human | *Helicobacter pylori* |
|---|---|---|---|
| Acc (%) | 95.90 | 99.33 | 90.95 |
| Pre (%) | 98.34 | 99.11 | 91.00 |
| Rec (%) | 93.37 | 99.49 | 90.88 |
| Spe (%) | 98.43 | 99.18 | 91.02 |
| F1 (%) | 95.79 | 99.30 | 90.94 |
| MCC (%) | 91.91 | 98.65 | 81.89 |

respectively. The best scores in these tables are highlighted in bold. Metrics not specified by the researchers in the original studies were calculated by reproducing the complexity matrix of the study (*Albu, Bocicor & Czibula, 2023*; *An et al., 2019*; *Chen et al., 2020*, *2019*; *Du et al., 2017*; *Gao et al., 2023*; *Li et al., 2022*; *Song et al., 2018*; *Tran et al., 2024*; *Yu et al., 2020*).

As seen in Table 3, it is apparent that the introduced model surpasses existing methods on *Saccharomyces cerevisiae* dataset across various measures, including accuracy, precision, recall, specificity, F1-score, and MCC. The MFPIC model, which achieved the best value in the table with an MCC value of 91.91%, provided an average of 2.6% improvement in this criterion. The model also showed an average of 1.2% improvement for the F1 score with a value of 95.79%.

Similarly, the proposed model was shown to outperform benchmark models on the human dataset as seen in Table 4, while achieving the second-best score only for the recall criterion. The best value in the table was achieved with an MCC value of 98.65%, providing an average improvement of 3.7% in this criterion. The model also exhibited an average improvement of 2.0% for the F1 score with a value of 99.30%.

On the *Helicobacter pylori* dataset, the model, which achieved the second-best score for the recall criterion, was more successful than the existing methods in all other criteria, as seen in Table 5. With an MCC value of 81.89%, it achieved the best value in the table and also provided an average of 6.2% improvement in this criterion. It also exhibited an average of 2.5% improvement in this criterion with an F1-score of 90.94%.

These improvements are primarily attributed to the proposed novel feature set which includes SCT feature, which captures more complex sequence motifs that are likely to influence protein interactions. Furthermore, the AAPD feature enhanced the model's spatial awareness, allowing it to better predict interactions that are dependent on the relative positioning of amino acids within the protein structure. Additionally, when compared to deep learning models that utilize PSSM features, the proposed model still exhibited superior performance. The inclusion of AAindex-based features, which encapsulate a wide range of physicochemical properties, contributed to a more nuanced representation of the proteins, ultimately improving the model's predictive power.

The integration of diverse features in the proposed model has provided clarification of the nature of PPIs. First, the success of the SCT feature suggests that non-adjacent amino

**Table 3 Comparison results of MFPIC with benchmark models on the *Saccharomyces cerevisiae* PPI dataset.**

| Method | Acc (%) | Pre (%) | Rec (%) | Spe (%) | F1 (%) | MCC (%) |
|---|---|---|---|---|---|---|
| EResCNN (*Gao et al., 2023*) | 95.34 | 98.32 | 92.26 | 98.43 | 95.20 | 90.86 |
| StackPPI (*Chen et al., 2020*) | 94.64 | 96.33 | 92.81 | 96.46 | 94.54 | 89.34 |
| DeepPPI (*Du et al., 2017*) | 94.43 | 96.65 | 92.06 | 96.82 | 94.30 | 88.97 |
| LightGBM-PPI (*Chen et al., 2019*) | 95.07 | 97.82 | 92.21 | 97.94 | 94.93 | 90.30 |
| GTB-PPI (*Yu et al., 2020*) | 95.15 | 97.97 | 92.21 | 98.09 | 95.00 | 90.45 |
| MM-StackEns (*Albu, Bocicor & Czibula, 2023*) | 94.00 | 94.50 | 93.30 | 94.57 | 93.89 | 87.87 |
| MFPIC (Proposed) | 95.90 | 98.34 | 93.37 | 98.43 | 95.79 | 91.91 |

**Table 4 Comparison results of MFPIC with benchmark models on the human PPI dataset.**

| Method | Acc (%) | Pre (%) | Rec (%) | Spe (%) | F1 (%) | MCC (%) |
|---|---|---|---|---|---|---|
| EResCNN (*Gao et al., 2023*) | 98.61 | 97.56 | 98.65 | 97.75 | 98.11 | 97.23 |
| DF-PPI (*Tran et al., 2024*) | 99.30 | 98.88 | 99.67 | 98.98 | 99.27 | 98.60 |
| SDNN-PPI (*Li et al., 2022*) | 98.94 | 99.02 | 98.77 | 99.10 | 98.89 | 97.57 |
| GWORVM-BIG (*An et al., 2019*) | 94.56 | 93.08 | 95.55 | 93.50 | 94.30 | 89.51 |
| RPEC (*Song et al., 2018*) | 96.59 | 96.18 | 96.72 | 96.48 | 96.44 | 93.18 |
| MFPIC (Proposed) | 99.33 | 99.11 | 99.49 | 99.18 | 99.30 | 98.65 |

**Table 5 Comparison results of MFPIC with benchmark models on the *Helicobacter pylori* PPI dataset.**

| Method | Acc (%) | Pre (%) | Rec (%) | Spe (%) | F1 (%) | MCC (%) |
|---|---|---|---|---|---|---|
| EResCNN (*Gao et al., 2023*) | 87.89 | 87.84 | 87.96 | 87.86 | 87.90 | 75.81 |
| StackPPI (*Chen et al., 2020*) | 89.27 | 90.37 | 87.93 | 90.60 | 89.12 | 78.59 |
| DeepPPI (*Du et al., 2017*) | 86.23 | 84.32 | 89.44 | 83.33 | 86.79 | 72.63 |
| LightGBM-PPI (*Chen et al., 2019*) | 89.03 | 88.36 | 89.99 | 88.13 | 89.16 | 78.14 |
| GTB-PPI (*Yu et al., 2020*) | 90.47 | 89.99 | 91.15 | 89.85 | 90.56 | 81.00 |
| MFPIC (Proposed) | 90.95 | 91.00 | 90.88 | 91.02 | 90.94 | 81.89 |

acid interactions are a critical component in determining protein interaction potential. This finding aligns with existing biological knowledge that emphasizes the importance of spatially close but sequentially distant residues in protein folding and function.

The AAPD feature further complements this by capturing the spatial relationships between amino acids in a sequence, which is crucial for understanding the three-dimensional conformation of proteins. This spatial information, combined with the evolutionary insights provided by PSSM, allows the model to make more informed predictions about protein interactions.

Moreover, the model's potential to outperform traditional and deep learning-based methods highlights the importance of integrating multiple features that capture both

sequence and structural information. This multi-faceted framework not only boosts prediction accuracy but also yields a more in-depth insight into the of the factors driving PPIs.

The proposed model's robust performance across different organisms, from yeast to humans, underscores its generalizability and potential applicability in diverse biological contexts. By accurately predicting PPIs, the model offers a valuable resource in various research domains, including drug discovery, where understanding protein interactions is critical for identifying therapeutic targets.

MFPIC integrates novel features such as SCT and AAPD, which capture spatial and structural information that traditional sequence-based features may overlook. This enables MFPIC to better model complex protein-protein interactions, particularly those involving indirect or non-local interactions.

While LightGBM-PPI and GTB-PPI employ tree-based ensemble methods optimized for tabular data, MFPIC uses a deep learning framework with batch normalization, dropout regularization, and adaptive learning rate scheduling. These techniques enhance the model's ability to generalize and handle non-linear relationships in high-dimensional feature spaces.

LightGBM-PPI and GTB-PPI models primarily rely on traditional features such as PSSM and AAindex-derived properties. Although these features are effective, the inclusion of SCT and AAPD in MFPIC provides richer and more comprehensive representations of protein sequences, leading to superior classification accuracy and robustness.

## Ablation analysis of feature contributions

To assess the significance of individual feature groups in the proposed model, an ablation analysis was conducted. In this analysis, specific feature groups were systematically removed from the dataset, and the model was retrained to evaluate the impact on performance metrics such as accuracy, precision, recall, F1-score, and MCC. The analyzed feature groups included SCT, AAPD, and Hydropathy.

The results of the ablation analysis on the *Helicobacter pylori* dataset are summarized in Table 6. Removing SCT features resulted in a noticeable decline in performance metrics, with the accuracy dropping to 80.14% and the MCC to 60.30% compared to the baseline model. This decline underscores the critical role of SCT features in capturing sequence-specific patterns that are essential for protein-protein interaction (PPI) prediction. The relatively larger decrease in recall (79.36%) suggests that SCT features play a significant role in identifying true positive interactions, thereby enhancing the model's sensitivity.

Similarly, the exclusion of AAPD features also led to a performance decline, albeit less pronounced. The accuracy dropped to 80.93%, and the MCC decreased to 61.87%. The results demonstrate that AAPD features contribute valuable spatial insights that enhance the model's ability to distinguish between interacting and non-interacting protein pairs.

The removal of the hydropathy feature had an important impact on performance. The accuracy decreased to 78.64%, and the MCC dropped to 57.29%.

These findings underscore the importance of combining SCT, AAPD, and hydropathy features in the proposed model. The combined use of these features allows the model to

**Table 6 Results of the ablation analysis for different encoding schemes for *Helicobacter pylori* dataset.**

| Removed features | Accuracy (%) | Precision (%) | Recall (%) | F1 score (%) | MCC (%) |
|---|---|---|---|---|---|
| SCT | 80.14 | 80.63 | 79.36 | 79.99 | 60.30 |
| AAPD | 80.93 | 81.41 | 80.18 | 80.79 | 61.87 |
| Hydropathy | 78.64 | 79.34 | 77.44 | 78.38 | 57.29 |

leverage sequence, spatial, and physicochemical information, resulting in robust PPI predictions.

## DISCUSSION

Although the proposed model shows high performance, the results are highly dependent on the dataset on which the model is trained. The model cannot guarantee the same performance on datasets belonging to different biological organisms. In addition to the class imbalance in the dataset, performance degradation can be observed especially in small and underrepresented subclasses.

In the analyses performed, it was seen that FP and FN values limit the model performance in some cases. Especially in biological systems, false negatives can lead to critical information loss, while false positives can cause biological experiments to be misdirected. The sensitivity of the model to misclassifications can lead to the model not always meeting the biological accuracy requirements.

Using additional feature extraction methods such as SCT and AAPD increases the computational cost. This cost increase was tolerated in order to increase the classification success. The performance of the model is dependent on the determined hyperparameter settings. Although hyperparameter optimization has been done extensively, it may need to be redone for different datasets.

## CONCLUSIONS

In this study, a novel approach for predicting PPIs is presented by integrating enhanced features, including SCT and AAPD, with existing methods. The proposed model demonstrated superior performance across multiple datasets, achieving higher accuracy and better generalizability in comparison to state-of-the-art methods.

The SCT feature, which considers non-adjacent amino acid interactions, was shown to be particularly effective in capturing complex sequence motifs associated with protein interactions. Similarly, the AAPD feature provided crucial spatial information that enhanced the model's understanding of protein structure and function. By combining these novel features with traditional approaches such as PSSM and AAindex-based features, the model was able to achieve a comprehensive representation of protein sequences, leading to improved prediction accuracy.

The findings of this research underscore the significance of incorporating diverse and complementary features in computational models for PPI prediction. The proposed model not only advances our understanding of the factors that drive PPIs but also provides a

valuable resource for researchers in the domains of bioinformatics and molecular biology. Ultimately, the proposed model signifies a remarkable breakthrough in the field of PPI prediction, with promising applications in various domains of biological research and drug development.

### Funding
The authors received no funding for this work.

### Competing Interests
The authors declare that they have no competing interests.

### Author Contributions
- Yunus Emre Göktepe conceived and designed the experiments, performed the experiments, analyzed the data, performed the computation work, prepared figures and/or tables, authored or reviewed drafts of the article, and approved the final draft.

### Data Availability
The data and computational model are available at GitHub and Zenodo:

- https://github.com/yegoktepe/predictPPI.

- Göktepe, Y. E. (2025). protein-protein interactions. Zenodo. https://doi.org/10.5281/zenodo.14744925.

The AAindex data (*Saccharomyces cerevisiae*, Helicobackter Pylori, Human) is available at: https://www.genome.jp/aaindex.

### Supplemental Information
Supplemental information for this article can be found online at http://dx.doi.org/10.7717/peerj-cs.2748#supplemental-information.

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
