# Peer review of "Protein-protein interaction prediction using enhanced features with spaced conjoint triad and amino acid pairwise distance"

_PeerJ Computer Science, doi:10.7717/peerj-cs.2748_

## Round 0.1 · original submission · Major Revisions

The reviewers have substantial concerns about this manuscript. The authors should provide point-to-point responses to address all the concerns and provide a revised manuscript with the revised parts being marked in different color.

Reviewer 1 ·

Basic reporting

The authors construct a comprehensive computational model, MFPIC, to predict protein-protein interactions by integrating diverse sequence-based features. Although experimental techniques for PPI detection have advanced, their limitations—such as high false-positive rates and resource intensity—necessitate robust computational approaches. However, I have several major concerns regarding the results and explanations provided, as outlined below:

1. The introduction of SCT and AAPD as novel features is commendable. However, further justification or real-world examples highlighting their unique contributions compared to traditional methods like PSSM is needed for better context.

2. While the manuscript benchmarks MFPIC against existing models, it would benefit from a discussion on why certain benchmarks (e.g., LightGBM-PPI, GTB-PPI) perform differently and what specific elements of MFPIC lead to its superior performance.

3. While metrics like accuracy, precision, and MCC are reported, a deeper analysis of false positives and false negatives could provide insights into the model's limitations.

Experimental design

1. The manuscript mentions cross-validation but lacks specifics about the splits (e.g., ratio, fold count). Providing these details ensures transparency in model evaluation and reproducibility.

Validity of the findings

1. The manuscript lacks a thorough discussion of the limitations of the proposed model.

2. Although the manuscript mentions that the source code and datasets are available on GitHub, there is no clear documentation or well-structured code provided.

Reviewer 2 ·

Basic reporting

1. The authors thoroughly review the relevant work on PPI prediction. It is recommended that the authors provide a more detailed classification of these methods and appropriately evaluate them to highlight the limitations of existing approaches.

2. The authors use multiple benchmark datasets to evaluate the proposed model. It is suggested that the authors include a table displaying the details of these datasets.

3. The flowchart of the model is not sufficiently clear. It is recommended that the authors add more descriptive content in the figure caption to improve the manuscript’s readability.

Experimental design

1. The manuscript compares the proposed model with different state-of-the-art methods on various datasets. However, the proposed model is a general approach for PPI prediction, it would be more appropriate to compare its performance with the same state-of-the-art methods across all benchmark datasets.

Validity of the findings

1. The authors construct protein features using various properties and encoding schemes. However, the contribution of each encoding scheme is unclear. The authors should include experimental results to demonstrate the contribution of different encoding methods.

2. An essential ablation analysis is lacking. The authors should indicate the contribution of different experimental settings or components to the prediction results.

Additional comments

1. Some recently published papers dedicated to the prediction of PPIs were missed in related work. A more comprehensive literature review should be presented.

---

## Round 0.2 · accepted · Accept

Reviewers are satisfied with the revisions, and I concur to recommend accepting this manuscript.

Reviewer 1 ·

Basic reporting

No further comment. Thank you for your kind responses, and I am happy that we can now have almost the same understanding of your research.

Experimental design

No further comment.

Validity of the findings

No further comment.

Reviewer 2 ·

Basic reporting

The quality of this article meets the journal's standard.

Experimental design

The experimental design of this article meets the journal's standard.

Validity of the findings

The findings have been properly validated.